# Sixty-Month Follow Up of Clinical MRONJ Cases Treated with CGF and Piezosurgery

**DOI:** 10.3390/bioengineering10070863

**Published:** 2023-07-20

**Authors:** Gianna Dipalma, Angelo Michele Inchingolo, Giuseppina Malcangi, Irene Ferrara, Fabio Viapiano, Anna Netti, Assunta Patano, Ciro Gargiulo Isacco, Alessio Danilo Inchingolo, Francesco Inchingolo

**Affiliations:** Department of Interdisciplinary Medicine, School of Medicine, University of Bari “Aldo Moro”, 70124 Bari, Italy; angeloinchingolo@gmail.com (A.M.I.); giuseppinamalcangi@libero.it (G.M.); ire.ferra3@gmail.com (I.F.); viapianofabio96@gmail.com (F.V.); annanetti@inwind.it (A.N.); assuntapatano@gmail.com (A.P.); dr.ciroisacco@gmail.com (C.G.I.); ad.inchingolo@libero.it (A.D.I.)

**Keywords:** MRONJ, osteonecrosis, denosumab, DRONJ, concentrated growth factor, platelet-rich plasma, platelet-rich fibrin, bone resection, oral surgery, piezosurgery

## Abstract

Aims: Medication-related osteonecrosis of the jaw (MRONJ) is a drug-related adverse reaction characterized by bone destruction and necrosis in the jaw. This case series aims to evaluate the treatment approaches and outcomes in MRONJ patients. Materials and methods: The retrospective study was conducted at the Dental Unit of the University of Bari, Italy. Patients with MRONJ were treated and followed up for 60 months. The treatment approach involved piezosurgery and concentrated growth factor (CGF). Six clinical cases from this group are described in detail. Results: None of the patients showed recurrence of necrotic MRONJ lesions during the follow-up period. The surgical interventions, including bone resections and the application of CGF, resulted in successful mucosal healing and the prevention of disease progression. Conclusions: This study highlights the complexity of managing MRONJ and the importance of a multidisciplinary approach. Conservative treatment options and minimally invasive surgery have shown efficacy in controlling symptoms and improving patients’ quality of life. However, the optimal treatment approach remains a challenge, and further studies are needed to evaluate alternative therapies and resective surgery. A comprehensive preoperative evaluation and collaboration among dental, endocrinology, and oncology specialists are crucial for personalized and multidisciplinary management. Ongoing research efforts are necessary to explore new therapeutic modalities and improve our understanding of MRONJ management, providing better support to patients dealing with this complex condition.

## 1. Introduction

Identified in the dental literature since 2003 [1], medication-related osteonecrosis of the jaw (MRONJ) is defined as a “drug-related adverse reaction characterized by progressive destruction and necrosis of the mandibular and/or maxillary bone of individuals exposed to treatment with drugs for which an increased risk of disease is established, in the absence of prior radiation treatment” [2,3].

MRONJ pathogenesis is based on the activity of medications that target osteoblasts, which are involved in bone remodeling, turnover, and vascularization. Osteoblasts deposit in the mineralized matrix and convert into osteocytes, which live for about 180 days [4,5]. They release osteoprotegerin (OPG), a protein that suppresses RANK and stimulates osteoclasts, reducing bone resorption [6,7].

When the osteocyte decays, OPG is no longer produced and, as a result of RANK-receptor binding on the osteoclast, necrotic or dysfunctional bone tissue is resorbed. This mechanism is the basis of bone homeostasis, which guarantees the elasticity and load-bearing characteristics of the skeletal structure [8,9].

MRONJ is a pathology that primarily affects the jaw bones because they have the following distinctive features:Bone turnover in the maxilla is 10 times faster than in the long bones, especially in the alveolar process of the post-extraction alveoli, the postero-lingual area, the maxillary sinus, and the torus [10];The mandibular vascularization is terminal;The mucosa and underlying periosteum are intrinsically exposed to trauma during masticatory phases;There is a high concentration of bacteria in the salivary biofilm;The periodontal ligament is also present [6,8,11].

The risk level is determined by the kind of medication administered, the dose, the frequency, and the length of administration, and the timing of the last dose. 

Several drugs have been identified in recent research as being associated with MRONJ [12]. These drugs include:Bisphosphonates (BPs): BPs are pyrophosphate analogues that form strong bonds with hydroxyapatite, which is the mineral component of bone. They are commonly used for the treatment of osteoporosis, bone metastases, and other bone-related conditions. However, long-term use of BPs has been linked to an increased risk of developing MRONJ [13].Tyrosine kinase inhibitors (such as sunitinib): these drugs are used in the treatment of various cancers, including kidney cancer and gastrointestinal stromal tumors. They work by inhibiting the activity of specific enzymes involved in cell signaling pathways. While they have shown efficacy in cancer treatment, they have also been associated with the development of MRONJ [14].Monoclonal antibodies (such as denosumab): monoclonal antibodies are designed to target specific proteins or receptors involved in disease processes. Denosumab, for example, is a monoclonal antibody used for the treatment of osteoporosis and bone metastases. However, like the other drugs on this list, denosumab has been linked to an increased risk of MRONJ [15].Angiogenesis inhibitors (such as bevacizumab): these drugs inhibit the formation of new blood vessels, which can be beneficial in the treatment of cancer and other conditions. However, they can also interfere with the normal healing process of the jawbone, leading to the development of MRONJ [16].Fusion proteins (such as aflibercept): fusion proteins are created by combining different protein components to target specific molecules involved in disease processes. Aflibercept, for instance, is a fusion protein used in the treatment of certain cancers. However, its use has been associated with an increased risk of MRONJ [17].mTOR inhibitors (such as Everolimus): mTOR inhibitors are a class of drugs that inhibit the mammalian target of rapamycin (mTOR), a protein involved in cell growth and division. These drugs have shown efficacy in cancer treatment, but they have also been associated with an increased risk of MRONJ [18].Radiopharmaceuticals (such as radium-223): radiopharmaceuticals are drugs that contain radioactive substances used for diagnostic or therapeutic purposes. Radium-223, for example, is used in the treatment of metastatic prostate cancer. However, its use has been linked to an increased risk of MRONJ [19].Estrogen inhibitors (such as raloxifene): estrogen inhibitors are drugs used in the treatment of hormone-receptor-positive breast cancer and osteoporosis. Raloxifene, for instance, is an estrogen inhibitor that has been associated with an increased risk of MRONJ [20].Immunomodulators (such as methotrexate and corticosteroids): immunomodulators are drugs that modify the immune response. Methotrexate, a commonly used immunosuppressive drug, and corticosteroids, which have potent anti-inflammatory properties, have both been linked to an increased risk of MRONJ [21].

MRONJ mostly affects patients taking one or more of these drugs because they present oncological diseases, osteoporotic diseases, or osteometabolic diseases, subsequent to exposure to a local triggering factor [22].

The most common local factor is tooth extraction (about 61%); however, MRONJ can also be initiated by chronic inflammation caused by untreated periodontitis, bone biopsies, the clinical elongation of crowns, surgery on the bone, and the insertion of a dental implant, all of which lead to trauma and the need for bone turnover, which these drugs inhibit [23].

Many categories have been developed over the years to help surgeons to determine the best treatment method based on the severity of the clinical picture, the cost–benefit ratio, and the patients’ different comorbidities [24].

In 2014, Favia et al. presented the dimensional staging of MRONJ to appropriately assess treatment plans.

Stage 0: this stage is characterized by the presence of clinical symptoms and nonspecific radiological signs, but there is no exposure of the underlying bone.Stage 1: in this stage, there is exposed bone measuring less than 2 cm, with or without associated pain.Stage 2: here, the exposed bone measures between 2 and 4 cm, and there is pain that can be managed with non-steroidal anti-inflammatory drugs (NSAIDs).Stage 3: this is the most advanced stage, where the exposed bone measures more than 4 cm. The pain experienced in this stage is not responsive to NSAIDs. Additionally, complications such as fistulae (abnormal openings) or the involvement of the maxillary sinus or the inferior alveolar nerve may be present.

It is important to note that these stages provide a framework for assessing the severity of MRONJ and guiding treatment decisions. Each stage represents a progression in the disease, with Stage 3 being the most severe and complex [25]. The effective management of MRONJ requires a multidisciplinary approach, the close monitoring of symptoms, and tailored treatment strategies based on the specific stage and individual patient needs [26].

Originally, surgical therapy was prioritized; however, medical professionals are now encouraged to adopt more medical, non-surgical treatments at an early stage [21]. According to the current AAOMS protocols, the surgical approach is recommended in more advanced stages of the illness [21,27].

## 2. Materials and Methods

### 2.1. Patient Selection and Treatment Protocol

The study describes patients with a diagnosis of MRONJ who were treated at the Dental Unit of the University of Bari.

Our treatment protocol for MRONJ followed a multidisciplinary approach involving medical and surgical interventions. The initial treatment approach was non-surgical or conservative, focusing on the management of infection and pain. This included the use of antimicrobial mouth rinses, antibiotic therapy, and non-steroidal anti-inflammatory drugs (NSAIDs). The specific medications and the treatment duration were based on individual patient needs.

For patients who did not respond to conservative treatment or had advanced stages of MRONJ, surgical intervention was recommended. The surgical approach involved the use of piezosurgery, which is a minimally invasive technique for bone resection. The necrotic bone tissue was removed using piezosurgery, and debridement and sequestrectomy were performed as needed. Flap design and suturing techniques were employed to promote proper healing.

Autologous preparations derived from the patient’s blood, such as concentrated growth factor (CGF), were used to enhance bone healing and regeneration. CGF was applied during the surgical procedures to promote stable repair over time.

### 2.2. Clinical Assessment and Follow-Up

The clinical assessment of the MRONJ lesions included radiographic examinations, cone-beam computed tomography (CBCT), and orthopantomography (OPT). The severity of the lesions was classified according to Favia’s staging system, which categorizes MRONJ into four stages based on the extent of bone involvement and associated symptoms.

Patients were followed up for a period of 60 months to monitor the healing of the MRONJ lesions and assess for any recurrence. Follow-up evaluations included clinical examinations, radiographic imaging, and assessments of pain symptoms.

### 2.3. Results

The results of the study are presented as clinical case descriptions for six patients with MRONJ who underwent treatment at the Dental Unit of the University of Bari. The treatment approach, including medical therapy, surgical intervention, and the use of CGF, is described for each case. The clinical and radiographic outcomes at different time points are also presented to demonstrate the effectiveness of the treatment approach.

The current retrospective clinical cases were treated in complete conformity with ethical standards, including the World Medical Association Declaration of Helsinki and the extra requirements of Italian legislation, at the Dental Unit of the University of Bari (Italy). Additionally, the University of Bari in Italy designated the project as being free from the need for ethical assessment since it includes only minor risks and the use of available information that consists only of non-identifiable data about humans. A formal informed consent form was signed by the patients. Our research group collaborated with the Phan Chau Trinh University of Medicine to carry out this restrospective study.

## 3. Case Descriptions

Our research group evaluated the 60-month follow-up of patients treated for MRONJ lesions with piezosurgery and CGF. None of the patients showed recurrences of necrotic MRONJ lesions. In the following, we illustrate the treatment of six clinical cases belonging to this group.

### 3.1. Clinical Case 1

This case featured a 78-year-old female patient affected by MRONJ following the intravenous administration of BPs. Radiographically, there was the presence in the fourth quadrant of a voluminous stage III lesion, according to Favia’s classification, which resulted in the fracture of the mandibular corpus (Figure 1). The patient underwent three cycles of pre- and post-operative medical therapy with the administration of metronidazole and oral anti-inflammatories and intramuscular ceftriaxone. The surgical approach (Figure 2) involved the piezosurgery-based resection of a large portion of the mandible, which was reconstructed by applying a mandibular fixation plate (Figure 3). Good mucosal healing was achieved in 21 days (Figure 4) and the follow-up orthopanoramic radiographs taken six months after the intervention did not show any recurrence (Figure 5).

### 3.2. Clinical Case 2

A 74-year-old female patient who was affected by osteoporosis received intravenous BPs that caused MRONJ in areas 33–35. The margins and size of the lesion were assessed by radiographic examination and CBCT (Figure 6); clinically, the lesion, which was at stage II according to Favia’s classification, was characterized by a large area of intraoral bone exposure (Figure 7). The treatment involved full-thickness-flap surgical access and the removal of necrotic bone tissue with piezosurgery (Figure 8). Mucosal healing was achieved in seven days, and the sutures were removed (Figure 9). The radiographic follow-up at six months was performed using CBCT (Figure 10).

### 3.3. Clinical Case 3

A 77-year-old male patient who received BPs therapy developed a degree I lesion in the second quadrant, according to Favia’s classification (Figure 11). The three cycles of medical therapy in the form of ceftriaxone, metronidazole, and antinflammatory drugs resulted in a remission of the pain symptomology but did not reduce the size of the lesion. A full-thickness flap was then designed to expose the necrotic bone tissue and remove it through the use of piezosurgery (Figure 12 and Figure 13). Curettage and the adequate suture of the flap (Figure 14) promoted complete mucosal healing after 21 days (Figure 15).

### 3.4. Clinical Case 4

Following the use of oral BPs, a 75-year-old female patient developed MRONJ in the area of teeth 35–36. The lesion was discovered radiologically using orthopantomography. (OPT) (Figure 16).

Piezosurgery was used as the surgical method. A full-thickness flap was made to expose the lesion and increase intra-operative vision (Figure 17A). The lesion was delineated with the PL3 Mectron^®^ tip, and the bone plug was removed once more (Figure 17A,B).

CGF gel and a membrane were utilized to enhance bone repair (Figure 18).

Complete healing occurred after 14 days (Figure 19).

### 3.5. Clinical Case 5

A 72-year-old female patient was treated with BPs for breast carcinoma. She presented a voluminous grade III lesion in the first quadrant that also involved the maxillary sinus, as evidenced by CBCT (Figure 20). Following the debridement of the necrotic bone fragment with piezosurgery, the application of a membrane and a suture was performed to promote proper mucosal healing (Figure 21).

### 3.6. Clinical Case 6

A 74-year-old male patient, who had received intravenous therapy with BPs for 6 years, presented MRONJ in region 3.7; it was detectable both from radiographic examinations with orthopantomography (Figure 22) and from a clinical examination (Figure 23). He presented a grade I lesion that was first treated with 3 cycles of medical therapy (ceftriaxone, metronidazole, and anti-inflammatory drugs) and then with surgical excision using piezosurgery (Figure 24 and Figure 25).

## 4. Discussion

In the literature, there are two types of treatment for MRONJ: non-surgical or conservative and surgical [28].

During the initial phases of the disease, a conservative approach is recommended, primarily relying on medication to control the infection and alleviate pain, with the aim of stabilizing the clinical condition and impeding disease progression. Ruggiero et al. state that the objectives of non-surgical treatment for MRONJ, utilizing antimicrobial mouth rinses and antibiotics, are to stabilize the lesions or reduce their severity [21,26,29,30].

In patients with MRONJ, the use of 0.2% alcohol chlorhexidine mouthwashes is recommended during the acute phases of an oral infection and as prophylaxis in the surgical perioperative phases [31]. A maintenance antiseptic program with non-alcoholic chlorhexidine 0.12% (two rinses per day, one week/month) is only indicated in people with MRONJ who are unable to undergo surgery due to comorbidities or non-deferrable antineoplastic therapies. The goal is to reduce the establishment of bacterial resistance and the deleterious repercussions of long-term chlorhexidine treatment [27,32].

The occurrence of bacterial infections plays a fundamental role in the etiopathogenesis of MRONJ; therefore, controlling it with antibiotic therapy is thoroughly justified [33]. The protocol presented in the literature provides as a first choice the oral administration of antibiotic combinations of penicillin and metronidazole, from a minimum of 7 to a maximum of 14 days; in the case of penicillin allergies, alternative molecules (erythromycin, clindamycin, or ciprofloxacin) are administered [21,26,30,31].

The surgical approach is intended for advanced MRONJ or individuals who have rejected conservative therapy [21,27].

In stages 1 and 2 of the disease, MRONJ is characterized by the presence of regions of necrotic bone, which can be removed using minimally invasive surgical techniques. More advanced stages of the disease are treated using more advanced surgical techniques [34]. Debridement and sequestrectomy are two examples of minimally invasive surgery. Debridement, also referred to as bone curettage, is the total removal of necrotic bone tissue until the appearance of a bleeding bone surface [24]. It is used when a viable bone is close to a dead bone. The surgery can be undertaken while the patient is under local or general anesthesia, and the defect is frequently completely corrected by mobilizing a muco-periosteal flap [9].

The surgical technique for eliminating necrotic bone sequestration, a section gradually divided from the underlying healthy bone, is known as sequestectomy. The sequester frequently exfoliates spontaneously; in certain cases, surgery under loco-regional anesthetic or general anesthesia is required, depending on the degree of the process, the clinical state, and the patient’s compliance [35].

In drug-related ONJ, the main objective of surgical therapy is to be curative rather than palliative; this is achieved by completely removing the diseased tissue and leaving only healthy tissue, to enable stable recovery over time [9].

If the condition to be treated is limited in scope, surgery will be less intrusive and have a higher chance of success [36,37]

Since bone is the tissue that is the most immediately affected by drug-induced ONJ from its earliest stages, the complete removal of the afflicted bone tissue should resolve any clinical issues without the need to remove the accompanying soft tissue [38]. Full and stable healing is enabled by the presence of histologically viable bone tissue at the margin of the bone excision [39].

It is essential to recognize the healthy tissue surrounding the lesion with a high margin of safety in order to completely remove the pathological bone tissue [40].

There are two approaches used in actual clinical practice: the first is based only on the evaluation of the margins of an intraoperative resection, whereas the second employs radiological methods to identify the true quantity of diseased tissue prior to surgery. In MRONJ, assessing bone bleeding is still the most commonly used approach for intraoperative surgical margin detection [41].

The expression “resective surgery” refers to the excision of diseased bone down to healthy tissue in one piece [39,42].

Drug-related ONJ and all other types of osteonecrosis and osteomyelitis of the jaws do not have standardized bone resection margins, in contrast to oncological surgery [40]. Pre-operative TC and RM evaluations of the resection margins allow for the accurate identification of the surrounding normal bone tissue, which, if not revealed to be pathological on histological inspection, ensures complete and stable healing over time [36]. Resective surgery is classified into two types: marginal and segmental. Marginal resective surgery involves the excision of diseased tissue in its entirety while maintaining the anatomical integrity of the skeletal segment in question. Depending on the complexity of the procedure, clinical circumstances, and the patient’s cooperation, this surgery can be conducted under local or general anesthesia [43].

The en bloc (full-thickness) excision of a skeletal segment with the interruption of its anatomical continuity is referred to as segmental resective surgery [39,44].

The most frequent form of mandibular reconstructive surgery is a mandibulectomy. It always leads to a loss of symmetry and occlusion of the lower half of the face. After surgery, titanium reconstruction plates, anatomical mandibular replicas, or vascularized bone flaps can be used to restore missing bone [45]. When titanium plates or a mandibular prosthesis are used to heal the mandible rather than vascularized bone flaps, the hospitalization time and recovery to normal function are shortened [39].

An upper jaw maxillectomy, a segmental resective procedure, is typically identified by the vertical and horizontal expansions of the defect caused by the removal of aberrant tissue [46]. The removal of only the dento-alveolar process, with or without palate preservation, distinguishes partial maxillectomies from full maxillectomies, which involve the complete removal of all bone sides, including the orbital floor [39,47].

According to the research, individuals with MRONJ have significant rates of recurrence/dehiscence following surgical resection, which results in more hospital stays and subsequent operations [48]. As a result, numerous methods for improving the existing therapy have been tried, but they have not yet been used in clinics [49]. Examples include altering the flap design or using intraoperative imaging to guide bone surgery. Another potential strategy is the use of autologous preparations made from the patient’s blood, such as platelet concentrate products including platelet-rich plasma (PRP), platelet-rich fibrin (PRF), and concentrated growth factor (CGF), which enhance and speed up bone repair or regeneration by releasing significant amounts of growth factors crucial for bone biology [50,51]. As it has been established that autologous platelet concentrates (APC) play a role in bone and soft tissue regeneration, they are used in a variety of dental procedures [52,53]. By attracting leukocytes, increasing collagen formation, producing anti-inflammatory compounds, and starting vascular internal development, APC aids in the healing process [52]. It has been shown that the amounts and rates of GF release vary between CGF, PRF, and PRP: PRP promotes the faster delivery of GFs to the target site, even though employing PRF or CGF leads to a significant increase in GFs, compared to PRP [54,55]. Additionally, it is important to take into account the major disadvantage of the PRP procedure, namely, the spread of infectious diseases and coagulopathies [54]. Unlike PRP, PRF and CGF only call for centrifuged autologous blood, providing immunological biocompatibility [52,56]. PRP, PRF, and CGF have all been used therapeutically for a number of conditions, including the treatment of MRONJ, with highly encouraging results in numerous trials. Researchers believe that, by improving patients’ quality of life and reducing pain and the incidence of postoperative infections, APCs may aid in the treatment of osteonecrosis [57]. Many studies have suggested using PRP to treat osteonecrosis brought on by bisphosphonates; Curi et al. treated patients with MRONJ with surgical necrotic bone excision and PRP, finding that complete wound healing was achieved in the majority of patients and that the time needed to treat BRONJ was cut in half [53,58]. The use of anticoagulants has been demonstrated to interfere with platelet-mediated angiogenic and regenerative responses, despite the fact that PRP has been proposed as a first-generation platelet concentrate [59].

### 4.1. Challenges and Treatment Approaches

The management of MRONJ presents several challenges. Non-surgical or conservative approaches are recommended in the early stages of the disease to stabilize the clinical picture and limit disease progression. Conservative treatment focuses on infection and pain management by using antimicrobial mouth rinses, antibiotic therapy, and non-steroidal anti-inflammatory drugs (NSAIDs). Surgical intervention is typically recommended for patients who do not respond to conservative treatment or have advanced stages of MRONJ. Surgical approaches involve debridement, sequestrectomy, and bone resection using techniques such as piezosurgery. The choice of treatment approach depends on the severity of the disease and individual patient needs.

### 4.2. Efficacy and Success Rates

The efficacy and success rates of non-surgical and surgical treatments for MRONJ have been demonstrated in clinical cases. Non-surgical approaches aim to stabilize or downstage the lesions, while surgical interventions aim for the complete removal of necrotic bone tissue. The use of autologous preparations derived from the patient’s blood, such as concentrated growth factor (CGF), has shown promising results in promoting bone healing and regeneration.

### 4.3. Complications and Side Effects

Complications and side effects associated with MRONJ treatment include the risk of recurrence or dehiscence following surgical resection, postoperative infections, and the need for subsequent operations. It is crucial to carefully evaluate resection margins and ensure the complete removal of pathological bone tissue while preserving healthy tissue.

### 4.4. Impact on Quality of Life

MRONJ can significantly impact the quality of life of affected patients, causing pain, functional limitations, and aesthetic concerns. The effective management of MRONJ requires a multidisciplinary approach, the close monitoring of symptoms, and tailored treatment strategies based on the specific stage and individual patients’ needs.

## 5. Limits

The study on medication-related osteonecrosis of the jaw (MRONJ) presented in this article has several limitations that need to be considered. Firstly, the study design is a case series, which has inherent limitations in terms of generalizability and the establishment of causal relationships. Additionally, the study was conducted in a single dental unit, which may limit the representativeness of the patient population and the applicability of the treatment protocol to other settings.

Furthermore, the retrospective nature of the study introduces potential biases and limitations in terms of data collection and analysis. The absence of a control group makes it difficult to assess the comparative effectiveness of different treatment approaches or to draw definitive conclusions about the efficacy of the interventions used.

Moreover, the follow-up period of 60 months may not be sufficient to fully evaluate the long-term outcomes and recurrence rates of MRONJ. Longer-term follow-up studies are needed to assess the durability of the treatment effects and to identify any potential late complications or relapses.

Additionally, the article does not discuss potential confounding factors, such as comorbidities or concomitant medications, which may influence the outcomes of MRONJ treatments. These factors can introduce variability and limit our ability to attribute the observed outcomes solely to the interventions used in the study.

Further research, including well-designed randomized controlled trials with larger sample sizes and longer follow-up periods, is needed to establish more robust evidence and guidelines for the management of MRONJ.

## 6. Conclusions

In conclusion, this clinical case series of medication-related osteonecrosis of the jaw (MRONJ) caused by bisphosphonates and monoclonal drugs highlights the complexity and importance of effectively managing this debilitating condition. The available conservative treatment options, including topical therapies, antibiotics, and minimally invasive surgery, have demonstrated their efficacy in controlling symptoms, slowing down disease progression, and enhancing the quality of life of affected patients.

However, selecting the optimal treatment approach still poses a significant challenge due to the need for further studies to thoroughly evaluate the benefits and risks of certain alternative therapies, such as teriparatide therapy or resective surgery.

The significance of a comprehensive preoperative evaluation, which encompasses the analysis of necrotic lesions and resection margins, cannot be overstated, as it ensures complete and stable healing over time.

While our understandings of and treatments for MRONJ continue to evolve, it is crucial for clinicians to maintain close collaboration with specialists from diverse disciplines such as dentistry, endocrinology, and oncology. This interdisciplinary collaboration facilitates the development of personalized and multidisciplinary approaches that are tailored to the specific needs of each patient.

Lastly, ongoing research efforts are imperative to explore new therapeutic modalities and enhance our understanding of MRONJ management. By making such efforts, we can provide the best possible support to patients dealing with this complex and continuously evolving condition.

## Figures and Tables

**Figure 1 bioengineering-10-00863-f001:**
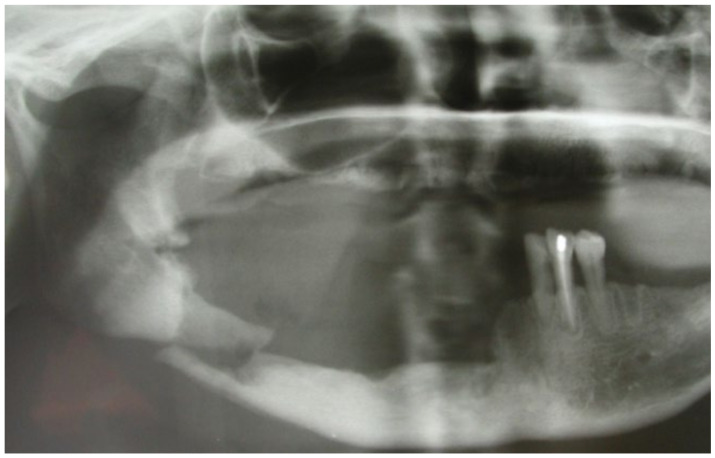
Pre-operative radiographic assessment of the lesion.

**Figure 2 bioengineering-10-00863-f002:**
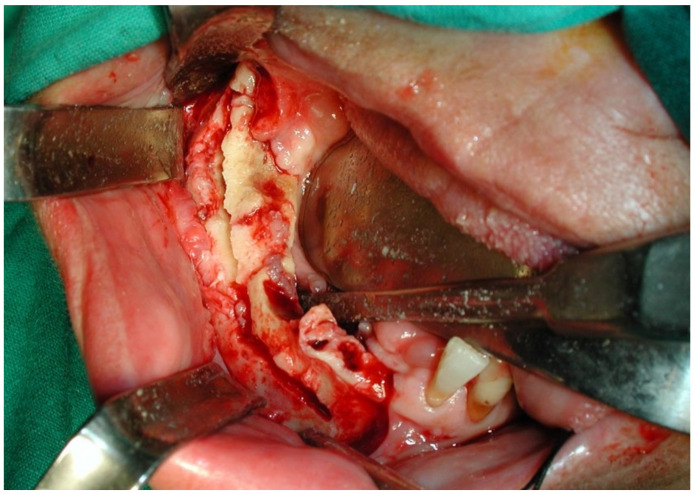
Intraoperative resection of necrotic bone tissue.

**Figure 3 bioengineering-10-00863-f003:**
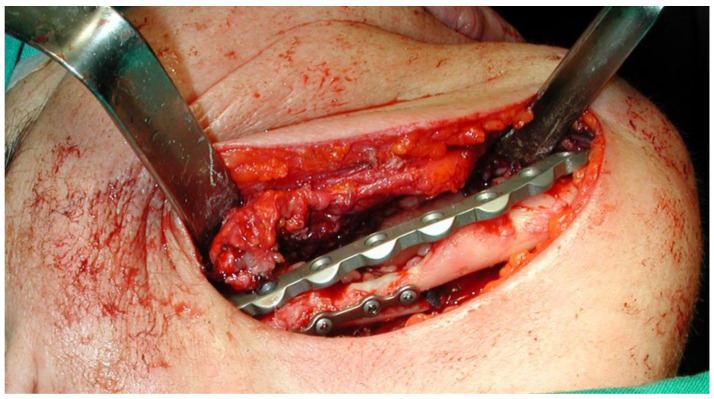
Application of the mandibular fixation plate.

**Figure 4 bioengineering-10-00863-f004:**
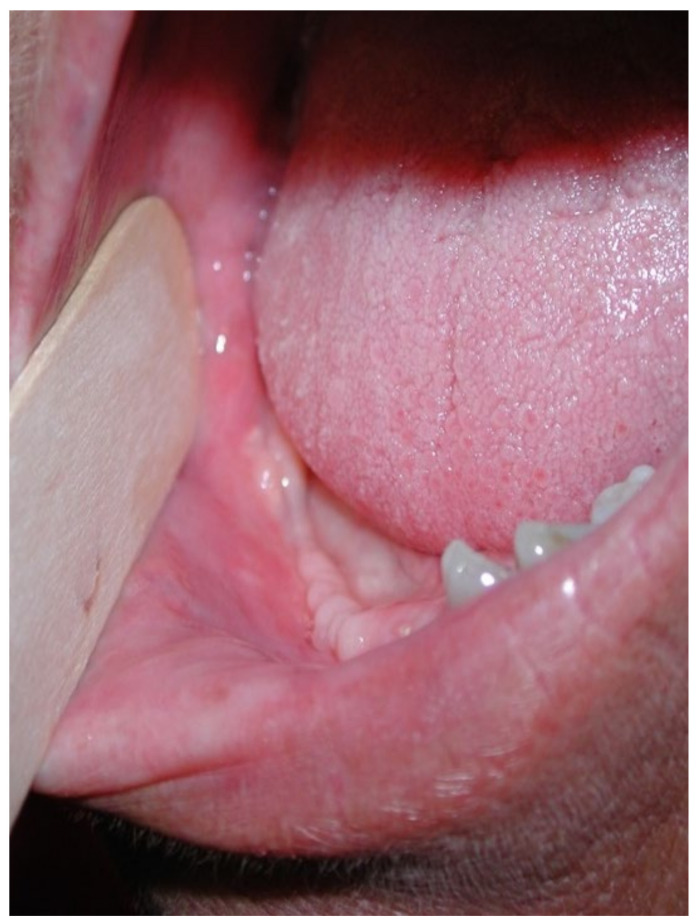
Follow-up at 21 days for Case 1.

**Figure 5 bioengineering-10-00863-f005:**
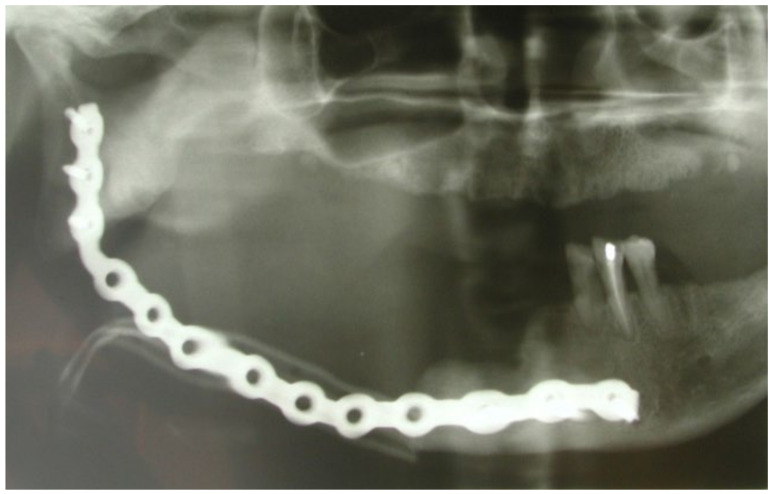
Post-operative radiographic evaluation.

**Figure 6 bioengineering-10-00863-f006:**
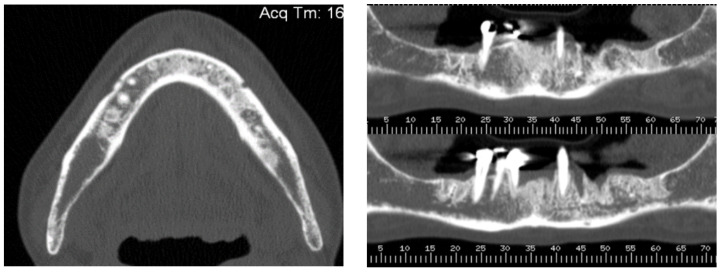
Pre-operative radiographic evaluation for Case 2.

**Figure 7 bioengineering-10-00863-f007:**
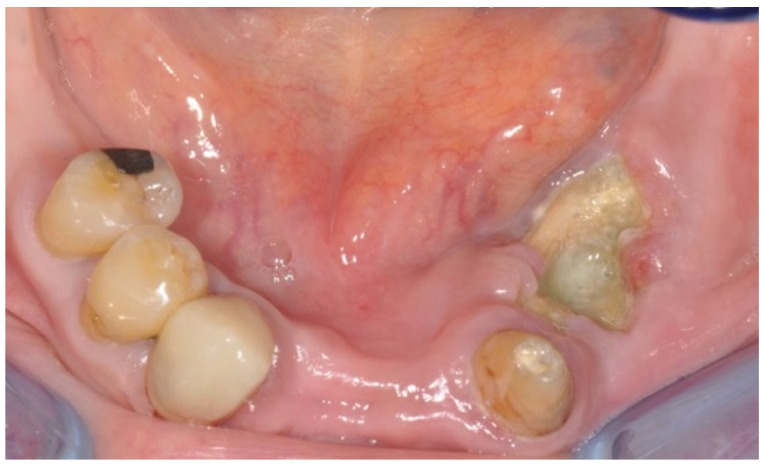
Intra-oral clinical evaluation of the lesion for Case 2.

**Figure 8 bioengineering-10-00863-f008:**
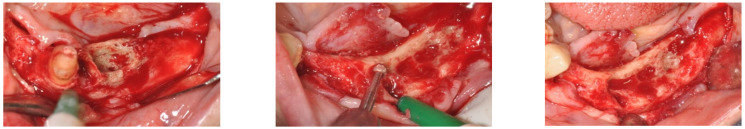
Full-thickness-flap surgical access and the removal of necrotic bone tissue with piezosurgery.

**Figure 9 bioengineering-10-00863-f009:**
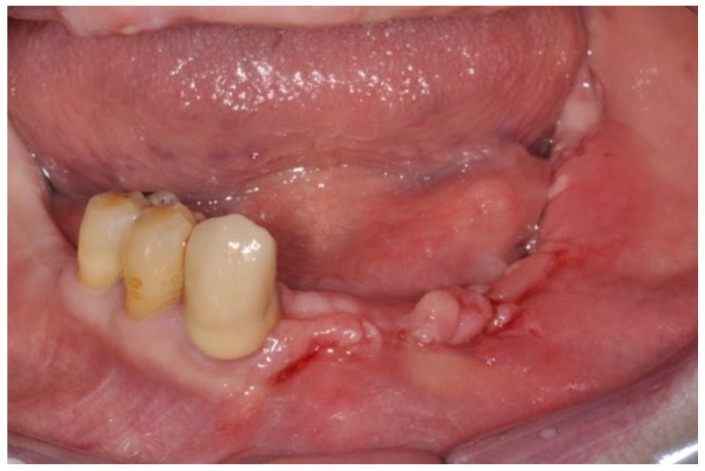
Seven-day follow-up.

**Figure 10 bioengineering-10-00863-f010:**
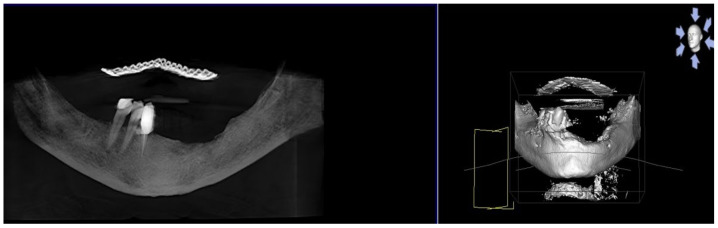
Postsurgical radiographic evaluation.

**Figure 11 bioengineering-10-00863-f011:**
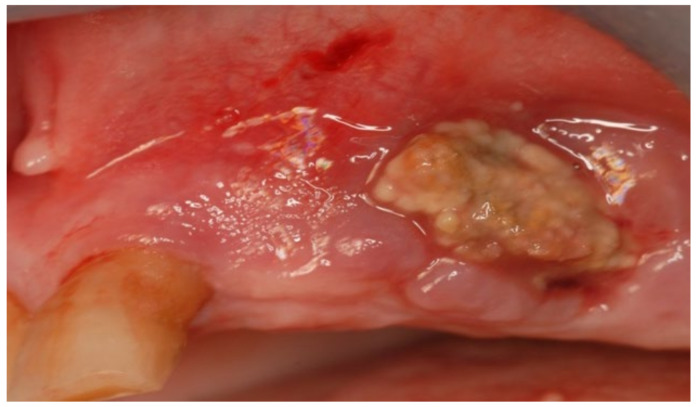
Intra-oral clinical evaluation of the lesion for Case 3.

**Figure 12 bioengineering-10-00863-f012:**
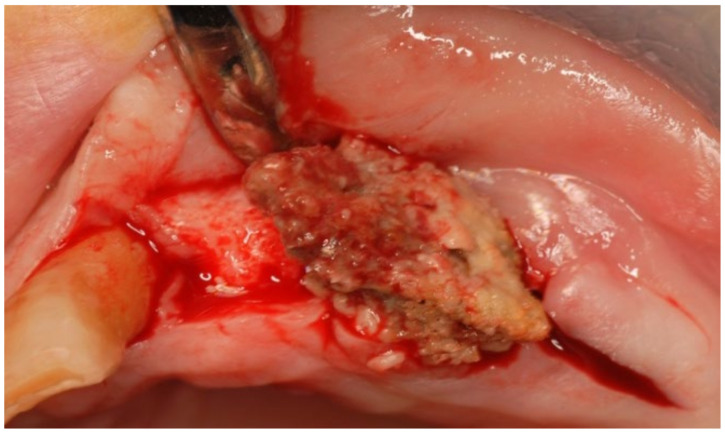
Surgical access and exposure of necrotic bone tissue.

**Figure 13 bioengineering-10-00863-f013:**
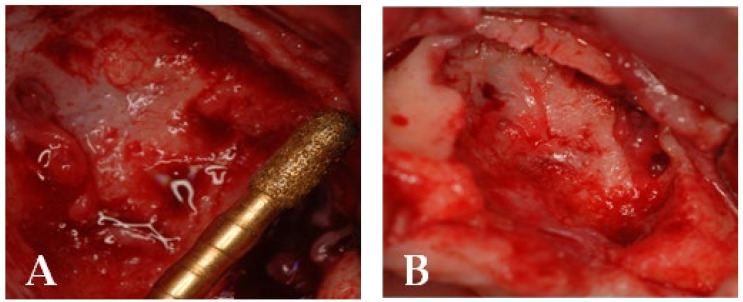
(**A**) Resection of necrotic bone tissue and curettage of the surgical site with piezosurgery. (**B**) Bone lesion after curettage with piezosurgery.

**Figure 14 bioengineering-10-00863-f014:**
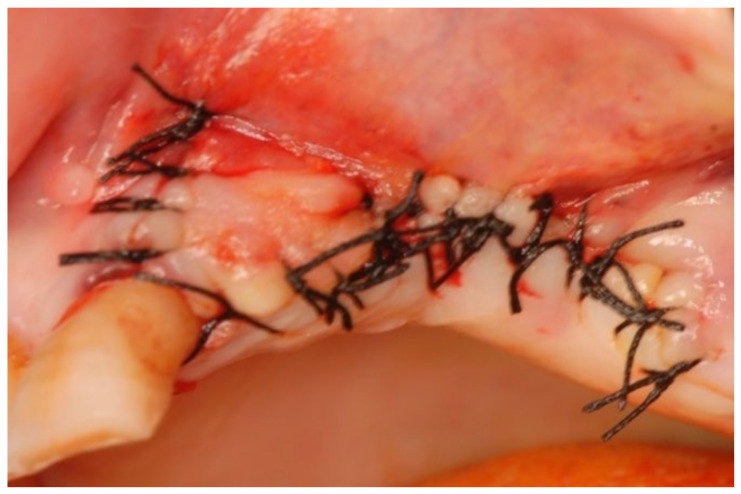
Suturing the surgical access flap.

**Figure 15 bioengineering-10-00863-f015:**
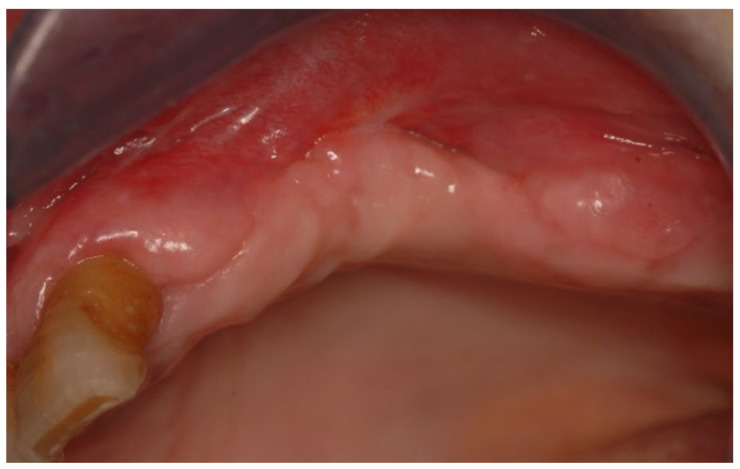
Follow-up at 21 days for Case 3.

**Figure 16 bioengineering-10-00863-f016:**
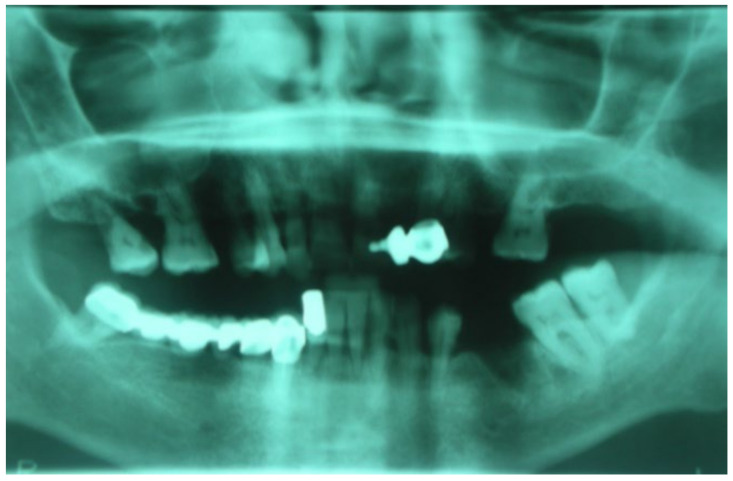
Initial lesion shown in OPT.

**Figure 17 bioengineering-10-00863-f017:**
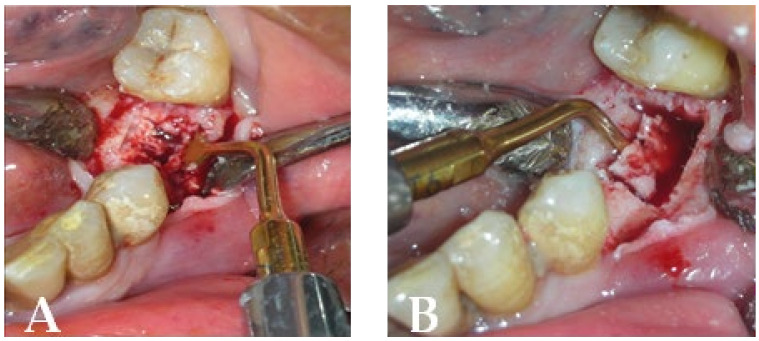
Intraoperative stage: (**A**) surgical area during necrotic tissue removal with piezosurgery; (**B**) bone plug remotion.

**Figure 18 bioengineering-10-00863-f018:**
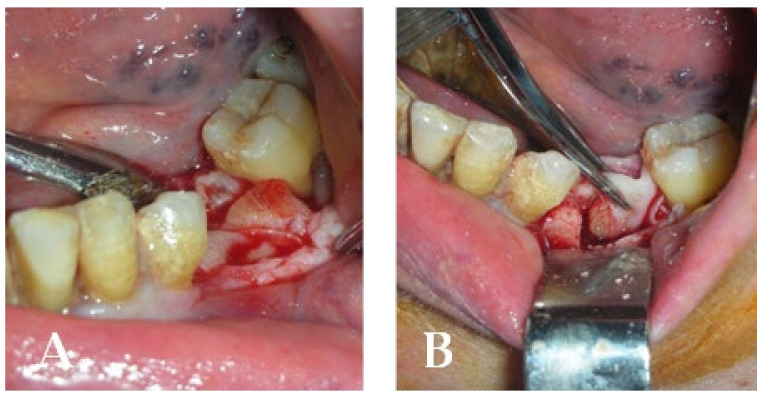
(**A**) Insertion into the residual cavity of CGF gel (concentrated growth factor); (**B**) residual cavity closure with the CGF membrane before suturing.

**Figure 19 bioengineering-10-00863-f019:**
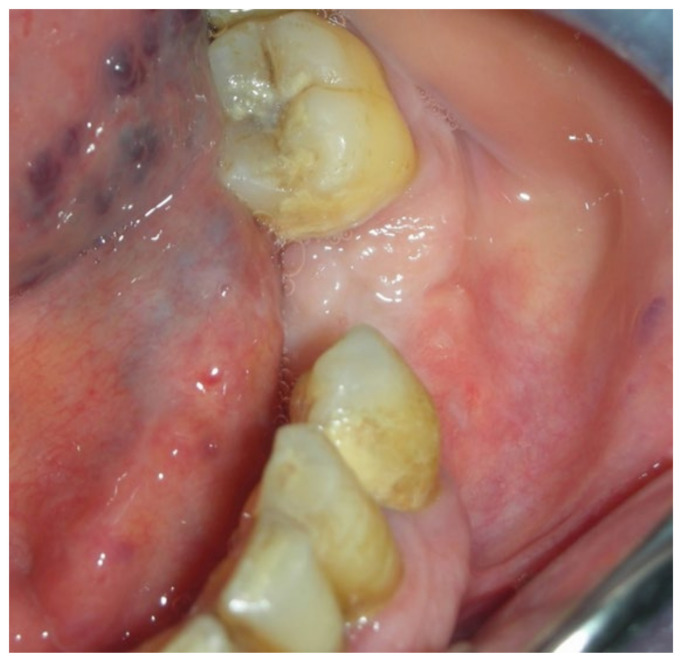
Complete healing 14 days after surgery.

**Figure 20 bioengineering-10-00863-f020:**
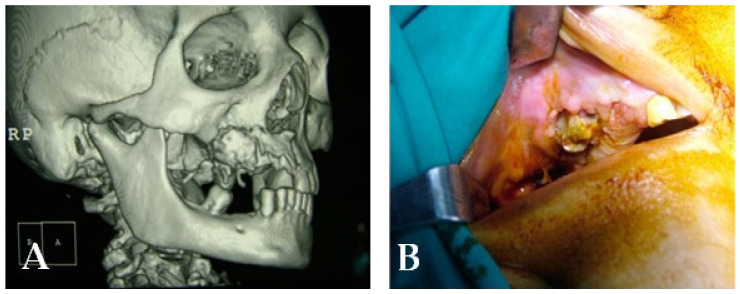
(**A**) Initial lesion shown in a CT scan. (**B**) Initial lesion.

**Figure 21 bioengineering-10-00863-f021:**
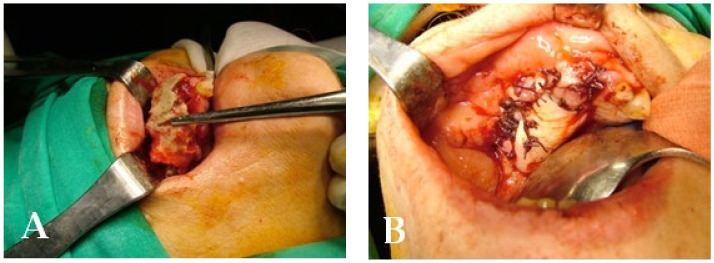
Intraoperative stage: (**A**) surgical area during necrotic tissue removal; (**B**) suture.

**Figure 22 bioengineering-10-00863-f022:**
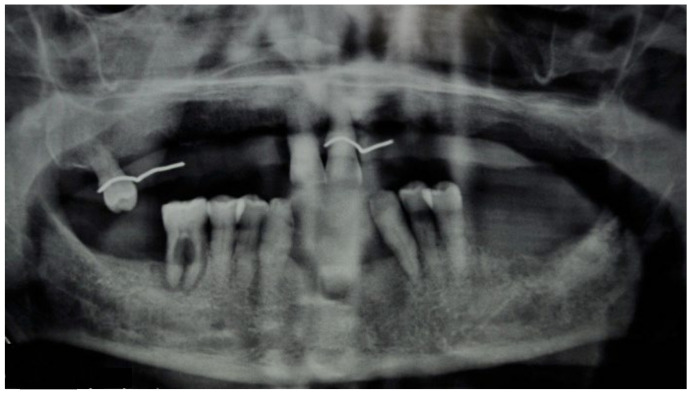
Pre-operative radiographic evaluation for Case 6.

**Figure 23 bioengineering-10-00863-f023:**
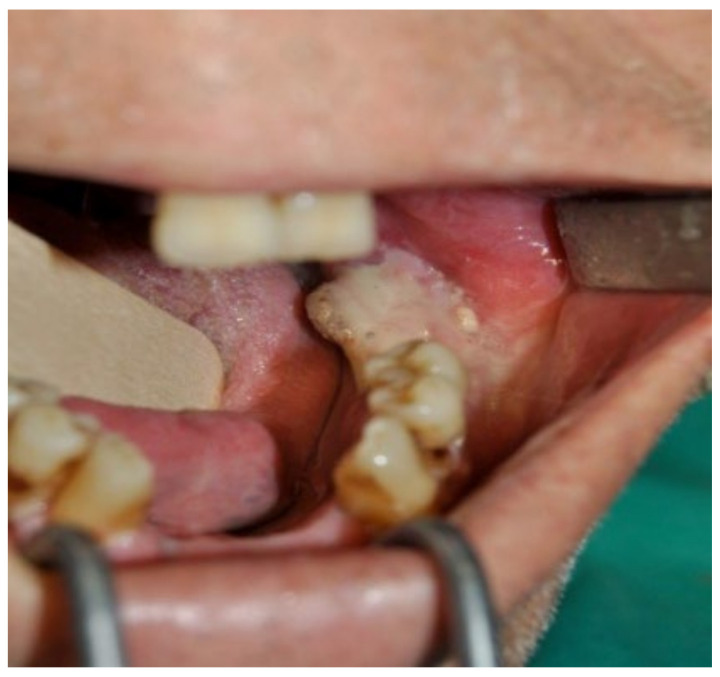
Pre-operative clinical evaluation.

**Figure 24 bioengineering-10-00863-f024:**
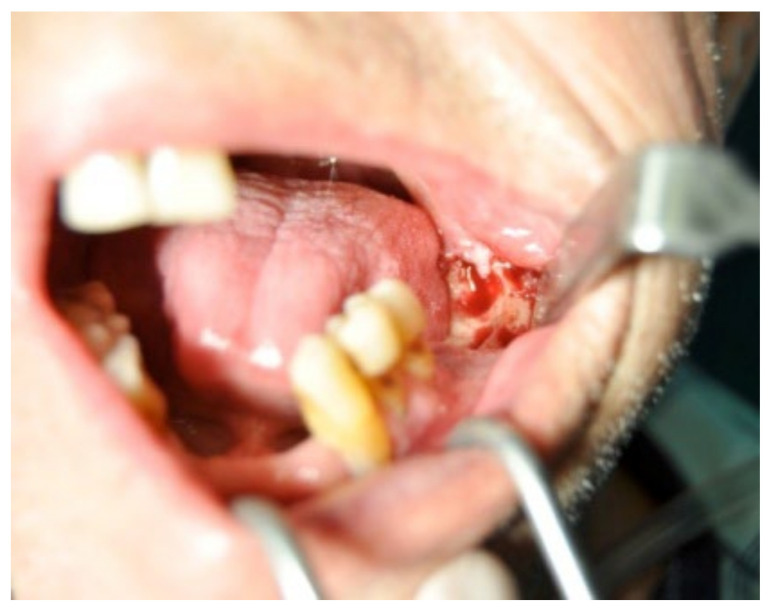
Surgical exposure of the necrotic bone.

**Figure 25 bioengineering-10-00863-f025:**
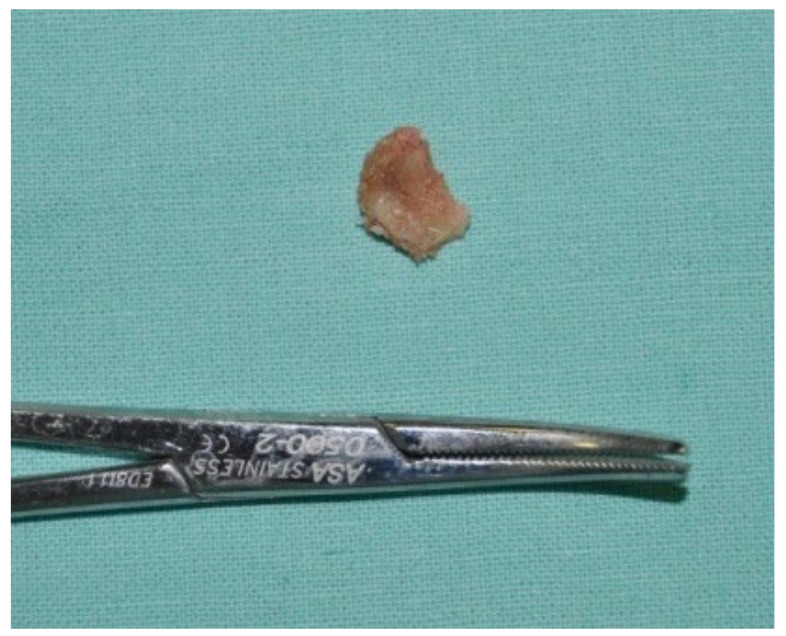
Surgically resected fragment of necrotic bone.

## Data Availability

Not available.

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
