# Peer review of "Sixty-Month Follow Up of Clinical MRONJ Cases Treated with CGF and Piezosurgery"

_bioengineering, 2023, doi:10.3390/bioengineering10070863_

Round 1
Reviewer 1 Report
Thank you for considering Journal of Maxillofacial Plastic and Reconstructive Surgery for the publication of your special case.
This is a well written manuscript, so this article can be accepted for publications.
Thank you for giving me your excellent journal's paper.
Reviewer 2 Report
Thank you for the opportunity to read this article. This is an ineresting article, but primarily it is unclear about the theme of this paper. You have to make it clear that this is case series, research, or review articles. It is very confusing. Under this uncler theme, I cannot agree with the publication. I think you need to revise as case sries, and revise abstract and all parts. 64 MRONJ clinical 2 cases is confusing. You can just show 8 cases as shown in this paper.
None
Reviewer 3 Report
Thank you for being entrusted with this task and for the chance to provide feedback that will hopefully contribute to the improvement of the manuscript.
Please write a structured abstract (including an objective, results…) not only describe what you will discuss. I also suggest avoiding abbreviations in the title.
Improve methodology and add a results section explaining the descriptive statistics of the study, reasons for including or excluding cases/patients, percentages…
The Discussion section lacks critical analysis and fails to address several important aspects. Firstly, it does not discuss the limitations or challenges associated with both non-surgical and surgical approaches. The efficacy and success rates of these treatments, as well as the potential complications or side effects, impact on the quality of life for affected patients are not thoroughly explored. Providing a balanced view of the advantages and disadvantages of each approach would have strengthened the discussion. Additionally, discussion lacks a paragraph with the limitations of the study.
There is too much in formation copied from this manuscript: https://www.mdpi.com/2076-3417/13/7/4370, please rewrite it.
In my opinion, the manuscript would greatly benefit from a thorough review of English grammar to ensure its accuracy. This will help to address the issues of clarity and comprehensiveness in order to enhance the overall quality of the document.
Round 2
Reviewer 2 Report
The authours properly revised to case series. I agree with the submission. This is an important case series.
Reviewer 3 Report
Thank you for revising the manuscript as requested. The authors have addressed the majority of the questions adequately, and with a few minor grammar corrections, I am confident that the manuscript is now prepared for publication.
The authors have addressed the majority of the questions adequately, and with a few minor grammar corrections, I am confident that the manuscript is now prepared for publication.